# THC and CBD Fingerprinting of an Elite Cannabis Collection from Iran: Quantifying Diversity to Underpin Future Cannabis Breeding

**DOI:** 10.3390/plants11010129

**Published:** 2022-01-04

**Authors:** Mahboubeh Mostafaei Dehnavi, Ali Ebadi, Afshin Peirovi, Gail Taylor, Seyed Alireza Salami

**Affiliations:** 1Department of Horticultural Sciences, Faculty of Engineering and Agricultural Science, University of Tehran, Karaj 31587-77871, Iran; mmostafaei@ut.ac.ir (M.M.D.); aebadi@ut.ac.ir (A.E.); 2CIAN Diagnostics, 5330 Spectrum Drive, Suite I, Frederick, MD 21703, USA; afshin700@yahoo.com; 3Department of Plant Sciences, University of California Davis, Davis, CA 95616, USA; 4Industrial and Medical Cannabis Research Institute (IMCRI), Tehran 14176-14411, Iran

**Keywords:** medical cannabis, hemp, marijuana, gene pool, landrace, phytocannabinoid

## Abstract

Cannabis (*Cannabis sativa* L.) has a rich history of human use, and the therapeutic importance of compounds produced by this species is recognized by the medical community. The active constituents of cannabis, collectively called cannabinoids, encompass hundreds of distinct molecules, the most well-characterized of which are tetrahydrocannabinol (THC) and cannabidiol (CBD), which have been used for centuries as recreational drugs and medicinal agents. As a first step to establish a cannabis breeding program, we initiated this study to describe the HPLC-measured quantity of THC and CBD biochemistry profiles of 161 feral pistillate cannabis plants from 20 geographical regions of Iran. Our data showed that Iran can be considered a new region of high potential for distribution of cannabis landraces with diverse THC and CBD content, predominantly falling into three groups, as Type I = THC-predominant, Type II = approximately equal proportions of THC and CBD (both CBD and THC in a ratio close to the unity), and Type III = CBD-predominant. Correlation analysis among two target cannabinoids and environmental and geographical variables indicated that both THC and CBD contents were strongly influenced by several environmental–geographical factors, such that THC and CBD contents were positively correlated with mean, min and max annual temperature and negatively correlated with latitude, elevation, and humidity. Additionally, a negative correlation was observed between THC and CBD concentrations, suggesting that further studies to unravel these genotype × environment interactions (G × E interactions) are warranted. The results of this study provide important pre-breeding information on a collection of cannabis that will underpin future breeding programs.

## 1. Introduction

*Cannabis sativa* L. (Cannabinaceae) is an annual herbaceous multi-purpose plant with a long history of human selective breeding [1,2,3,4,5].

The genus *Cannabis* contains different types of chemicals with a diverse phytocannabinoid profile and range of effects [1]. The differences in phytocannabinoids composition and quantities of cannabis chemotypes should be searched in the genetic background of their biosynthesis pathways and the environmental conditions where they have been evolved [6,7,8]. Precursor synthesis of cannabinoids occurs from two distinct biosynthesis pathways: the polyketide and the methylerythritol phosphate (MEP) pathways, which produce olivetolic acid (OLA) and geranyl diphosphate (GPP), respectively [5]. Geranylpyrophosphate:olivetolate geranyltransferase catalyse the alkylation of OLA with GPP, leading to formation of CBGA (cannabigerolic acid), the main precursor of various cannabinoids, responsible for producing acidic precursors of THC (tetrahydrocannabinolic acid; THCA) and CBD (cannabidiolic acid; CBDA). Naturally these phytocannabinoids exist as both monocarboxylic acids (e.g., THCA, CBDA) and as decarboxylated forms (e.g., THC, CBD); however, heating (at temperatures above 120 °C) promotes decarboxylation [9,10,11,12,13,14].

Different types and concentrations of the cannabinoids, in particular, THC with psychotropic effects and CBD, a non-intoxicating metabolite, alongside applied morphological attributes may underpin the recreational, medicinal, and industrial uses of cannabis [15,16,17,18,19,20,21,22,23]. The range of 0.3–1% THC that determines the border of drug-type and non-drug type cannabis of course seems to be only a widely accepted agreement to determine the restrictions of cultivation in different countries [24,25,26,27,28]. Nowadays, there is intense competition for finding unique chemotypes or varieties with low THC (<0.2–0.3% dry weight) and high CBD contents that can be industrially used or cultivated [10,29,30,31,32,33,34].

Despite high potential as a multipurpose plant to produce drugs, fiber-based products, nutritional supplements and seed oil, and cosmetics, there remain critical gaps in knowledge. The recent discoveries of a variety of medical cannabis with different preparations to treat and or cure a number of serious disorders and its newly discovered industrial applications have given momentum to the quest for exploring, exploiting, and protecting natural resources with commercial potential [35,36,37,38].

The quantity and composition of cannabinoids, in particular, THC and CBD, have been targeted by extensive research by breeders, the scientific community, and legislative authorities for almost half a century [5,8,39].

Here, we considered that concentration and the ratio of THC/CBD could be the crucial key points for the initiation of a breeding program towards different end products in the huge cannabis/hemp market. Although cannabis has a long history of cultural use in Iran, there is little detailed information about the Iranian cannabis market, including the chemical characteristics of the locally available cannabis landraces, which are stored in the CGRC gene bank. To complete the puzzle, the current study was conducted as the primary report of screening THC and CBD contents of 20 selected native cannabis populations collected from different locations in Iran towards initiating anew breeding program for different industrial and medical purposes.

## 2. Results

### 2.1. THC and CBD Quantification and Phenotyping

The study of chemical patterns and profiling of novel cannabis germplasm grown in highly diverse climates and geographical regions are likely to valuable and informative as a starting point for future cannabis breeding programs. HPLC-derived chromatograms of the extracts from the female buds and standards solution are shown in Appendix A.

Variation of THC and CBD composition of cannabis plants derived from 20 locations indicate that this collection is diverse in THC and CBD both within and between populations, and that phenotypes characterized by THC/CBD values could be classified into three classes: Type I (THC-predominant), Type II (approximately equal proportions of THC and CBD), and Type III (CBD-predominant). Furthermore, studied populations were completely different in terms of many morphophysiological features, which is beyond the scope of this study, but the plants of three unique dwarf and early flowering populations with compact inflorescence including Pir-01, Sqz-01, and Naq-01 were all located chemically in the group of Type I. Additionally, the plants of location Ard-01, which was morphologically distinct in terms of inflorescence structure, seeding time, and seed size, surprisingly, it failed to classify into either THC-predominant or CBD-predominant groups. The majority of plants of this population with almost equal concentration of THC and CBD, hereafter, a ratio of THC/CBD≈ 1, were defined as Type II. The different structure of female inflorescences and buds in studied populations is presented in Figure 1.

Our results illustrated that maximum THC was observed in the plants collected from locations Dez-02 and Dez-01 distributed in the south-west areas. The minimum THC was quantified from plants from the location Sam-01, the maximum CBD was obtained from the plants of location Rmhz-01, and the minimum CBD was obtained from Pir-01 (Figure 2 and Figure 3a). A significant variation in THC content from 0.02% to 15.66% and CBD from 0.03% to 15.89% (% DW) were observed in this collection (Figure 2). The distribution of THC and CBD concentration (%) for each location is shown in Figure 2. A greater THC/CBD amount can be recorded due to high THC or low CBD levels. The populations of this collection with a prevalence of THC and THC/CBD content well beyond 1.0, including Pir-01, San-01, Sqz-01, Zah-01, Qzv-01, Naq-01, Abh-01, Krmn-01, Krsh-01, Bsh-01, Bam-01, Dez-01, and Dez-02, are defined as Type I. Additionally, six populations, including Nhv-01, Ban-01, Sir-01, Rmhz-01, Ark-01, and Sam-01, with a prevalence of CBD, hereafter, THC/CBD well below 1.0, were considered CBD- predominant populations [8,29,30,31,32,33,40]. Among CBD-predominant populations, all plants of population Sam-01 were characterized with THC level ≤0.3%. The phenotype of each population based on THC/CBD ratio is given in Figure 3a. As we observed THC and CBD diversity both between and within studied populations, so did we measure the abundance (%) of each phenotype (Type I, Type II, and Type III) per population in terms of the ratio of THC/CBD (Figure 3b). In order to define the phenotype of a population, we have considered the phenotype that is dominant in that population (Figure 3a); then, we have shown the abundance of all existed phenotypes in that population as well (Figure 3b). All plants of population Ard-01 had approximately equal proportions of THC and CBD, and three populations of Ban-01, Ark-01 and Sam-01 were 100% CBD- predominant and all their individuals assigned to the group of Type III (Figure 3b).

### 2.2. Correlation between the Climatic and Geographical Parameters and THC and CBD Quantity Variation

We considered all 13 parameters including statistics of latitude, longitude, elevation, P, Tmean, Tmin, Tmax, RHmean, RHmin, RHmax, SD, WS, DP (abbreviations are explained in detail in the Materials and Methods section “climatic and geographical data”), and THC and CBD concentration (%) obtained for the plants of each location to estimate the Pearson correlation coefficient among these variables. Our results showed that there was a negative correlation between THC and CBD concentration, but this was not significant (r = −0.053). Latitude had a negative significant correlation with both THC and CBD (r = −0.157, P ≤ 0.05; r = −0.239, P ≤ 0.01, respectively). In addition, elevation correlated negatively with THC and CBD, and this was significant (r = −0.408, P ≤ 0.01; r = −0.222, P ≤ 0.01, respectively).

Moreover, we observed a significant negative correlation for wind and THC concentration (r = −0.326, P ≤ 0.01). All three statistics related to temperature, including Tmean, Tmin, and Tmax, showed a positive correlation with THC (r = 0.352, P ≤ 0.01, r = 0.292, P ≤ 0.01, r = 0.412, P ≤ 0.01, respectively; Table 1), as well as CBD content (r = 0.314, P ≤ 0.01, r = 0.311, P ≤ 0.01, r = 0.287, P ≤ 0.01, respectively; Table 1). Additionally, all three relative humidity statistics comprising RHmean, RHmin, and RHmax correlated negatively with CBD content (r = −0.201, P ≤ 0.05, r = −0.171, P ≤ 0.05, r = −0.188, P ≤ 0.05, respectively), and RHmin correlated negatively with THC content (r = −0.206, P ≤ 0.01). In addition, to assess the correlation of each parameter, we performed PCA on THC and CBD content of all studied individuals and climatic and geographical variables of sampled regions (Figure 4). The PCA plot revealed a negative correlation for elevation and confirmed the positive correlations of the three temperature statistics including Tmean, Tmin, and Tmax with THC and CBD content (Figure 4).

## 3. Discussion

### 3.1. THC and CBD Diversity

In this study, we have found that Iran is a rich natural resource of cannabis with a high level of THC and CBD diversity, both among and within populations, and this can be attributed to genetic diversity, age, nutrition, geographical, and bioclimatic factors influencing plant chemistry [41,42,43].

The greatest THC variation within locations was obtained for plants from locations of Dez-02 and Dez-01 and the greatest CBD variation was found for plants in Rmhz-01 (Figure 2). In addition, the phenotype diversity was found within locations, so that all three types of I, II, and III with various abundances were found within six locations of San-01, Zah-01, Qzv-01, Krmn-01, Krsh-01, and Bam-01 (Figure 3b). The diversity within these populations can help genetic erosion reduction [44].

All individuals of three populations of Ban-01, Ark-01, and Sam-01 belonged to the type III (CBD- predominant), and all plants of Ard-01 assigned to Type II (roughly equal proportions of THC and CBD), indicating that these populations are more chemically homogeneous compared to other populations.

### 3.2. Morphological Features and Phenotyping

The populations under study differed in morphological features including total height (tall, mid, or dwarf), thin or squat growth, leaf shape, phyllotaxy, number of nodes, number of lateral inflorescences, internode length, compact or non-compact inflorescence, flowering time (fast, mid, or late), seeding time (fast, mid, or late), seed features, etc., and although not reported here, this is in line with earlier research reported that chemical phenotypes can be characterized by different morphological features [8]. Danziger and colleagues (2021) reported architecture of cannabis plant may considerably affect the cannabinoids profile, which has significant pharmaceutical and economic importance [45]. Additionally, phenotype markers that can facilitate preliminary identification and selection as a supplement to chemical and genetic analysis developed in 2021 [46]. They showed significant morphological differences in terms of leaf color, leaflet shape, large and compact inflorescences, and dense and resinous trichomes, which were identified between 21 cultivars covering three chemical phenotypes (THC dominant, CBD dominant and intermediate cultivars). They also reported that modern cannabis cultivars are morphologically distinguished by a morphological feature (leaflet shapes) by users and breeders [46].

Among the studied populations, three dwarf populations including Saq-01, Pir-01, and Naq-01 distributed in the north-west along with 10 other populations were located in the group of Type I (Figure 3a). These three unique populations were assumed to be marijuana, as they have similar features to those reported recently with thinner stems, more branches, and a higher density of floral tissues than industrial hemp plants [8]. In addition, the plants of population of Ard-01 were morphologically (tall, branched with space between branches, branches out and few leaves, early seeds mature, and drop) and chemically (located in the group of Type II) distinct.

### 3.3. Phenotyping in Terms of THC and CBD Composition

Cannabis is an economically important species and is predicted to become a significant commercial crop with unprecedented market growth potential [47]. Recent publication showed among individual cannabis plant parts including roots, leaves, stem bark and inflorescence, cannabis inflorescence was characterized by the highest concentration of cannabinoids in three chemovars [48]. They stated that the comprehensive profile of bioactive metabolites can rediscover therapeutic potential for each part of cannabis from their traditional use [48]. Chemical screening of natural populations can help identify chemical diversity, which is a primary step for improving breeding programs in this plant.

Taken together, the cannabis definitions are different based on scientific and political assignations. The significant difference in cannabinoid content of cannabis is supported by numerous studies showing that the most important classification of cannabis types that vary widely among political jurisdictions is that of the drug type and the fiber type (hemp). THC is the major cannabinoid in marijuana types, while CBD predominates in fiber-type hemps [3,10,32,33,49,50,51,52]. The cultivation of cannabis varieties containing up to 0.2% ∆-9-THC with no indication of permitted percentage of the other compounds, first of all being non-addictive psychoactive cannabidiol (CBD), have been recently allowed by two European regulations and Italian law. These varieties have been used for food, oils, fiber, powder, and bioengineering [41]. Recent studies on chemical composition and quantification of hemp industrial varieties indicated ∆9-THC content was lower compared to other cannabinoids. Additionally, ∆8-THC was detected only in one hemp oil sample at too low a concentration [53]. Additionally, a report in 2019 indicated that, among several cannabinoids, only CBDA was determined to show a different concentration in hemp inflorescences samples. Additionally, according to their study, THCV was not found in the hemp inflorescence samples analyzed, and ∆-9-THC and ∆-8-THC were detected at low concentrations, below the legal limit. Therefore, their results confirmed the classification of the studied samples as fiber [54]. In order to grow consumer interest in hemp oilseed supplements, four main cannabinoids of CBD, CBDA, CBN, and ∆-9-THC in an oil matrix of seven commercial hemp oil supplements have been determined [55]. They reported that the cannabinoid composition is required to be monitored in such supplements, as in some cases, the cannabinoids concentration in analyzed samples differed significantly from those declared by the manufacturers [55].

In addition, cannabis chemotypes have been reported using the biochemical composition, in particular, the THC/CBD ratio in many publications. Chemical phenotypes (chemotypes) can be used to define cannabis varieties with different chemical variants and different morphological features to classify *C. sativa* into three principal classes differ in their THC/CBD ratios: chemotype I (drug type plants which exhibit a THC/CBD ratio well beyond 1.0 especially due to high THC content); chemotype II (an intermediate ratio of 0.5–2.0); and chemotype III (fiber-type plants that have a ratio well below 1.0, with a prevalence of CBD) [8]. Additionally, cannabis varieties were classified into three groups: chemotype I (total THC/total CBD ratio ≥ 1.0), intermediate type (chemotype II), which has an intermediate ratio close to 1.0, and chemotype III, which exhibit a low total THC/total CBD ratio ≤ 1.0 [3]. On the other hand, three main classes of cannabis have been suggested based on THC/CBD ratio: THC-type plants with THC/CBD ≥ 10, intermediate-type plants with THC/CBD ≈ 1, and CBD-type plants with THC/CBD ≤ 0.1 [56,57]. Marchei and colleagues (2020) stated that, while the THC content in light cannabis has to be within 0.2%, CBD content is highly variable, ranging from 2 to 40% [41]. Additionally, serum THC/CBD concentration ratio was used as a useful biomarker to identify use of light cannabis (not higher than 0.9), illegal THC cannabis (generally >10), and medical cannabis (>1) [35].

Another study on chemotaxonomic discrimination indicated significant chemical differences in three chemotypes, so that CBD dominant varieties had higher amounts of total CBD, while THC dominant varieties had higher total THC, and intermediate varieties were generally equal to or in between those in CBD-dominant and THC-dominant varieties [58]. They finally showed that chemotype markers (presence or absence of THC and CBD) could be used as chemical fingerprints for quality standardization or variety identification for clinical studies and cannabis product manufacturing [58]. THC and CBD variations among populations of this Iranian collection enabled us to define studied populations as three different groups: Type I (THC/CBD > 1), Type II (THC/CBD ≈ 1), and Type III (THC/CBD < 1) with a prevalence of THC, both THC and CBD in an approximately equal proportions, and CBD, respectively (Figure 3a).

Genetic diversity in Iranian cannabis germplasm has been assessed by merging the data with the marijuana and hemp data prepared by [59] to elucidate the relationship of Iranian cannabis with marijuana and fiber type accessions. Finally, they categorized Iranian cannabis populations into marijuana and hemp clusters and reported that natural populations of cannabis in Iran in general more closely fit the profile of marijuana than hemp [60]. Additionally, in this study, we have used the same sources to fingerprint THC and CBD, and these populations were defined as Type I, Type II, and Type III, thus revealing, on the basis of chemistry, these three distinct types.

In this research, most plants of population Ard-01, which were also morphologically distinct with a THC/CBD ratio around one due to equivalent THC and CBD concentrations, were assigned to class Type II, defined as an intermediate THC/CBD ratio. This supports the findings of a genetic diversity study reported earlier, indicating Ard-01 was failed to group with either the marijuana or hemp clusters [60]. This is an interesting finding and may have immediate significance for commercialization, promising for therapeutic purposes in the production of medications with formula requirement of THC/CBD ratio around the unity such as Sativex^®^ [61]. This is an important medication for the suppression of spasticity and pain associated with multiple sclerosis [13,62]. It is worth noting that sequencing the whole genome of this morphologically, chemically, and genetically distinct population would be of value, as this should provide further insight into the genetic basis of the three chemotypes described here.

Furthermore, a concentration of THC ≤ 0.3% and higher amount of CBD in the plants from the location Sam-01 that was located in class Type III is another significant finding in this study, worthy of further investigation, and may reflect a tight control of cannabinoid type and content for cultivation and could provide pre-breeding germplasm resources for future development of hemp crops, particularly in nations where there is a strict regulatory environment around the production of high THC crops.

Although most of the focus has been on identifying plants with higher concentrations of THC for the recreational drug industry, and those with higher concentrations of CBD for medicinal, fiber, and grain purposes, there is also evidence reporting therapeutic benefits for CBD with anticonvulsive, anti-epileptic, antimicrobial, and antiparkinsonian properties, which are more important recently due to the lack of psychotropic effects associated with CBD consumption, as well as FDA-approved CBD drugs such as Epidiolex^®^ [63,64,65,66,67,68]. Berman and colleagues (2018) found that, despite the similarity in CBD contents, not all equally high-CBD cannabis extracts produced the same effects. They stated that, as cannabinoids profiling of diverse medical cannabis plants are different, analyzing the effects of specific cannabis compositions for pharmacological-based research is critical [69]. Therefore, it seems likely in the future that both THC and CBD content may be of wide relevance for further development within the pharmaceutical industry [68]. This is supported by a previous study reported that the combination of the psychoactive cannabis Δ9-THC with other non-psychotropic cannabinoids such as CBD demonstrated a higher activity than THC alone [70,71]. Additionally, previous findings showed that all three major products—food, fiber, and medicine—were extracted from the same crop of the accessions from Darchula district in the northwest of Nepal [72]. In addition, some Iranian cannabis populations were evaluated using wood and fiber anatomy and stem biometry characteristics [73]. They suggested that both populations of Bsh-01 and Zah-01 are significant candidates in terms of fiber anatomy, fiber length, and stem biometry and can be considered for textile and paper industries, while in this study, the aforesaid populations were defined as Type I and, despite the prevalence of THC, also have eligible fiber anatomy characteristics.

Our research showed that populations of Nhv-01, Ban-01, Sir-01, Rmhz-01, Ark-01, and Sam-01 are defined as Type III. However, in the earlier study, some populations are expected to be putative high-potential fiber populations, indicating that both populations Ban-01 and Nhv-01 have strong fiber characteristics such as a higher average of bast and woody cores and, alongside populations of Rmhz-01 and Ark-01, can be considered an option for breeding programs towards producing fiber [73].

The results of this study contribute important pre-breeding information for cannabis breeders to improve breeding programs utilizing this collection. Although the uses of these populations cannot be predicted with certainty, as accessions high in THC or high in CBD are Type I and III, respectively, they are not necessarily “hemp” or “medicinal marijuana” in the classic sense; however, according to previous study using genomic data, Iranian populations were located in two distinct marijuana and hemp clusters [60]. Therefore, populations of this collection assumed to offer a range of measurable health benefits in the pharmaceutical, dietary supplement industries, dual-proposal (seeds and fiber), and even renewable and sustainable feedstock for the production of biofuels [5,74,75,76,77]. Further studies are needed to evaluate the potential of these populations. Additionally, fiber anatomy is required to assess for fiber production purposes.

### 3.4. Effect of Climatic and Geographical Characteristics on THC and CBD Content Variation

In general, correlation analysis and PCA results revealed a positive correlation between temperature variables and the two target chemical metabolite contents, and a negative correlation between latitude as well as elevation and metabolites content (Table 1; Figure 4). It is clear that higher temperatures promote cannabinoid biosynthesis, as found in this research. We also found a negative correlation between THC and CBD concentrations, but it was not significant. The THC and CBD biosynthesis pathways have been elucidated and show that cannabigerolic acid (CBGA) is a prerequisite for both CBD and THC biosynthesis and then follows two pathways to synthesize carboxylic acids (THCA or CBDA), and these acidic forms of cannabinoids, upon heating or smoking, decarboxylate to their neutral forms (THC and CBD) [13,21]. THCA synthase and CBDA synthase enzymes that catalyze the reaction of THCA to THC and CBDA to CBD, respectively, compete with each other for CBGA and expedite neutral cannabinoids creation and their levels. The negatively correlation between latitude and THC and CBD contents supported by a study reported that latitude decreases can result in cannabinoids level reduction, and plants from high latitudes exhibited a low ∆9-THC [78].

Moreover, this study indicated that the populations at the high elevation showed a trend towards lower concentrations of THC and CBD. For example, Sam-01 at 1858 m above sea level (maximum value of elevation) had lower cannabinoids, and Dez01, Dez-02, and Bsh-01 at a low elevation had higher amounts of THC and CBD compared to other locations studied. The two populations located in the warmest state of Iran, including Dez-01 and Dez-02, had the highest concentrations of THC, consistent with the positive effect of temperature on cannabinoid accumulation and growth. Overall, these results showed that environment is also likely to play a role in cannabinoids concentration, suggesting that controlled environment studies or multi-year trials should be completed to further elucidate the importance of these G × E interactions.

## 4. Materials and Methods

### 4.1. Genetic Resources

Twenty natural dioecious cannabis populations selected from the native cannabis gene pools across Iran (CGRC; www.medcannabase.org). These populations were located in distinct geographical and climatic regions (included cold and mountainous regions of the west and north-west, warm and wet regions of the south-west, and warm and dry regions of the east and south-east areas) (Table 2; Figure 5, Appendix A). At least 10 female plants per population were used for phytochemical analysis if available (Table 2). In addition, seeds were collected from labeled local populations to grow for future studies. Geographical information was recorded and used to determine climatic parameters of sampled regions.

In order to minimize sampling bias, analyses were performed on female flower buds as the major sources of glandular trichomes, the main tissue for phytocannabinoids accumulation [8,9,11,32,50,57,79,80]. Due to a diverse phenological pattern among 20 populations, terminal inflorescences were collected before seed appearance, when trichomes turned 50–70% amber and the rest remained clear to capture the highest concentration of THC and CBD for quantitative analysis (Table 2).

### 4.2. Sample Preparation

Fresh plant materials were air dried at room temperature in darkness for up to 14 d until the leaves become brittle. The drying time varied depending on inflorescence density: plants with compact and tight buds took 14 d, whilst plants with branched buds with space between branches were dried after 7 d. At this stage, the water content of the plant materials was approximately 10%, which was uniform for all populations. Coarse dried female flower buds were then selected, crumbed, and pulverized until ensuring accepted tolerance homogeneity of the samples [81]. A mixture of methanol/chloroform (*v*/*v*: 9/1) was used as a cannabinoid’s extraction solution [3,82,83,84,85]. After testing different protocols, sonication was found to be the best process to agitate samples for cannabinoid extraction [86]. Thus, 50 mg of fine tissue powder was weighed and extracted with 2 mL of a mixture methanol/chloroform (*v*/*v*: 9/1) by sonication for 40 min and centrifugation at 10,000 rpm for 15 min at 10 °C. The upper phase was separated overnight to evaporate the solvent, and residue (dried extract) was dissolved in 1 mL of HPLC grade MeOH. In order to filter the extract, centrifugation was performed at 13,000 rpm for 10 min. Finally, the supernatant was transferred into an amber vial and 20 μL applied for injection to HPLC apparatus.

### 4.3. Chemicals

Cannabinoid reference standards for THC and CBD were purchased from Cerilliant (Sigma Aldrich, USD; TK#61-65 and TK#61-477, respectively), and all standards had purity of 100%. All analytical reagent-grade and HPLC-grade solvents used for the extraction procedure were purchased from Merck KGaA (Darmstadt, Germany). Ultrahigh-pure distilled and deionized water was prepared with the Millipore-Q water purification system from Millipore (Kloten, Switzerland).

### 4.4. THC and CBD Standard Preparation

Stock standard solutions containing THC and CBD at a concentration of 1.0 mg mL^−1^ in methanol were prepared. Working solutions at different concentrations were prepared by dilution of the stock standard solutions with methanol to create calibration curves with linear ranges. Calibrators containing THC and CBD were applied at concentrations of 7.8125, 15.625, 31.25, 62.5, 125, 250, 500, and 1000 μg mL^−1^ (*w*/*v*). Stocks were stored at −20 °C in amber sealed glass vials in the dark until analysis.

### 4.5. HPLC analysis of Cannabinoids

The contents of THC and CBD were measured by HPLC. All chromatographic runs were carried out using a Smartline model (Knauer, Berlin, Germany) system equipped with a Smartline pump 1000 (Knauer, Germany), Smartline Manager 5000 degasser 10 mL (Knauer, Germany), UV–VIS (DAD)- 2800 model, and reversed-phase column C18 column, which was protected by a KNAUER guard column and pre-column (MZ Cartridges 10 mm). Data and integration were processed by Software ChromGate (V 3.1.7). The separation was performed using an isocratic flow, and the mobile phase consisted of a mixture of methanol: water in the ratio 80:20 (*v*/*v*) as long as the composition stayed the same [22,87]. The injection volume was 20 μL, and injection was performed via HAMILTON, SYR 50 μL (750 N ga 22s/51 mm/pst3). Peaks were monitored at 220 nm [24,86,88]. Flow rate was set to 1 mL min^−1^. Stock standard solutions containing THC and CBD at a concentration of 1.0 mg mL^−1^ had been prepared in methanol diluted to an appropriate concentration range for construction of calibration curves. THC and CBD peaks were identified and quantified by congruent retention times compared with those of authentic standards obtained. The concentration of THC and CBD in samples were measured using calibration equations (yCBD = 67191x, r2 = 0.9948; yTHC = 61827x, r2 = 0.9976). Column temperature was set at 30 °C. The flowchart of cannabinoids quantification steps is presented in Figure 6.

### 4.6. Climatic and Geographical Data

Climatic data for each site of origin were acquired from the Iran Meteorological Organization (https://data.irimo.ir, accessed on 28 November 2019) (Appendix A). The data set contained a total of 13 variables, including latitude, longitude, elevation, and 10 bioclimatic variables representing average annual precipitation (P), average annual temperature (Tmean), average annual minimum temperature (Tmin), average annual maximum temperature (Tmax), average annual relative humidity (RHmean), average annual minimum relative humidity (RHmin), average annual maximum relative humidity (RHmax), average annual sunshine duration (SD), average annual wind speed (WS), and average annual dew point (DP). Multivariate statistical analysis including PCA was performed on these variables using SAS software, PRINCOMP procedure to reduce high dimensionality of variables and evaluate correlation among them.

### 4.7. Statistical Analysis

Box plot analysis was performed on Excel for THC and CBD diversity in each location using phytochemistry data of THC and CBD contents. The ratio of THC/CBD was used to show different chemotypes of studied populations. Additionally, IBM SPSS statistics software was applied to elucidate the correlation of studied target metabolites concentration and climate factors at site of origin. In other words, a Pearson correlation coefficient was used to determine the relationship between environmental/geographical variables and chemotype to explore the effects of these variables on increase or decrease in two target cannabinoids.

## 5. Conclusions

Starting a successful breeding program in cannabis using a new elite germplasm requires profiling of cannabinoids and terpenes in selected superior chemotypes that harbor ideal morphological characteristics for diverse needs to develop hybrid seeds. A deeper insight into the patterns of recreational, industrial, and medical cannabis use is a high priority for both public health and industry. So far, there is still a great lack of information about chemical composition and cannabinoids profile of Iranian cannabis populations in terms of THC and CBD contents. The present study as a first survey provides a deep insight into THC and CBD profile of 20 natural dioecious cannabis populations morphologically distinct from various geographical regions of Iran and the plausible correlation of these contents with environmental and geographical conditions of regions of origin. The results showed that diverse THC and CBD contents both between and within populations represented three chemical phenotypes as type I (THC-predominant; 13 populations), type II (approximately equal proportions of THC and CBD; 1 population), and type III (CBD- predominant; 6 populations). Ard-01 with THC/CBD ratio of 1 was chemically distinct, which may increase the capacity of commercialization and medical industries. The THC content of all plants of population Sam-01 characterized with THC level ≤ 0.3% contribute to increase the potential of law enforcement programs in Iran. Additionally, differences in unique and important morphological features of this collection may indicate the difference in their chemical type. Correlations between geography and climate of site of origin were also identified, suggesting that both THC and CBD production were positively and negatively correlated with temperature and latitude, respectively, but more research is required to tease apart these G × E interactions more fully. In conclusion, our study unravels the natural diversity to delineate cannabis resource with variations in THC and CBD contents and morphological traits, providing a foundation to initiate breeding programs in Iranian cannabis towards different industrial and medical purposes. Therefore, this study will promote future possibilities for the burgeoning cannabis industry in Iran.

## Figures and Tables

**Figure 1 plants-11-00129-f001:**
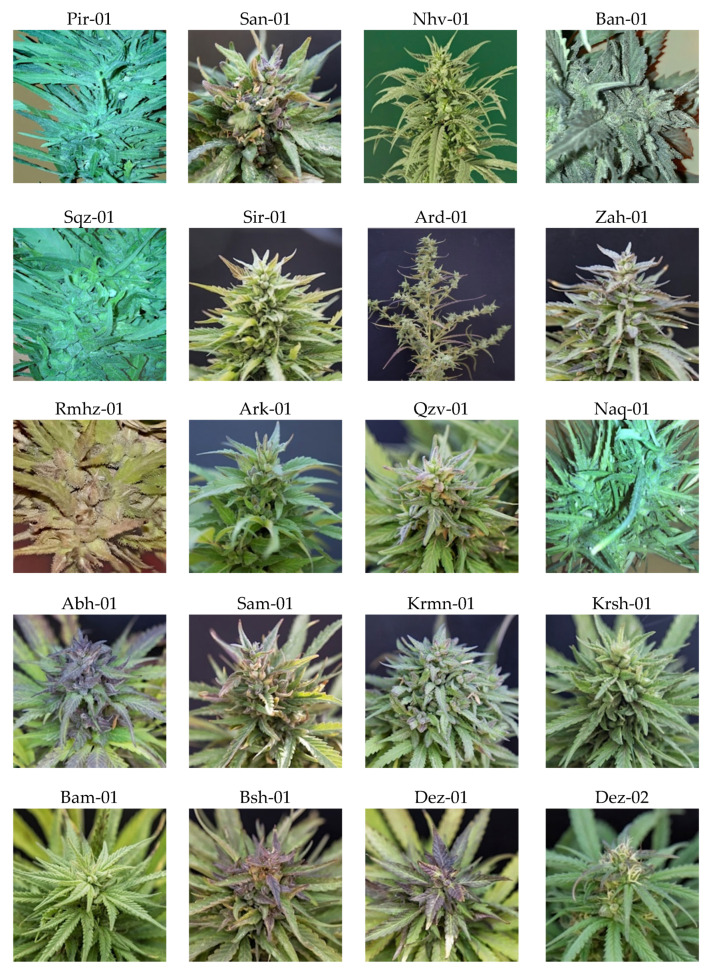
Diversity in female inflorescence of 20 natural cannabis populations distributed across Iran from west and north-west to south-west and east and south-east.

**Figure 2 plants-11-00129-f002:**
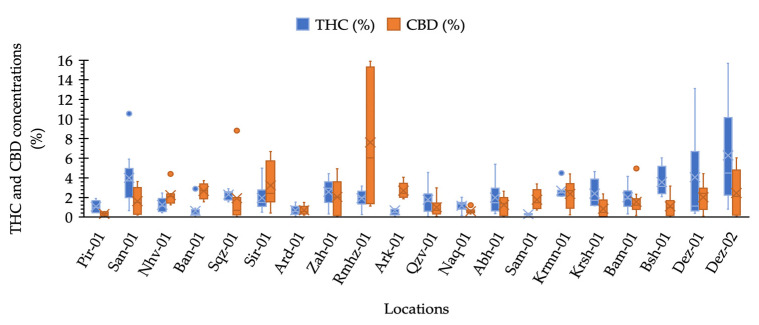
Box plot analysis of CBD and ∆-9-THC concentration (%) distribution measured for pistillate cannabis plants in each location.

**Figure 3 plants-11-00129-f003:**
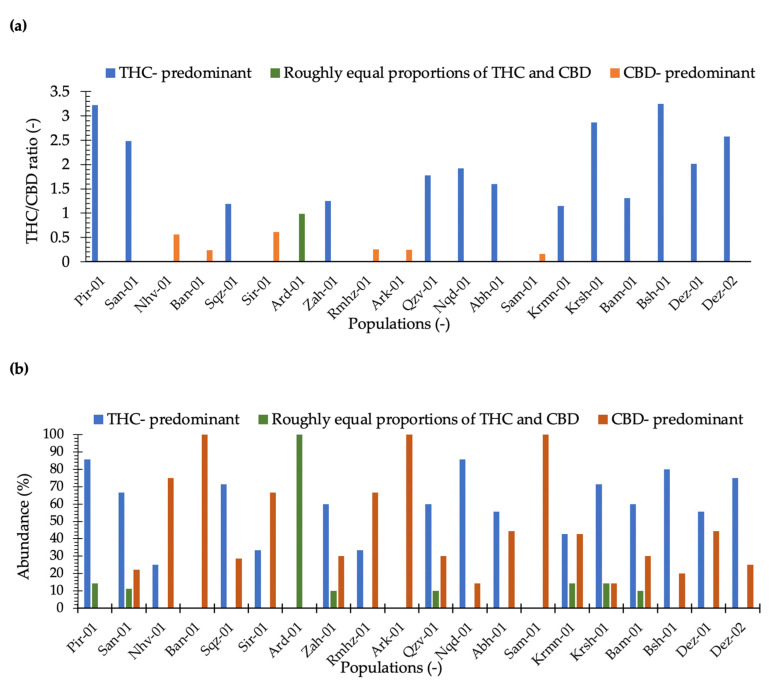
(**a**) THC/CBD ratio has been classified populations in three chemical phenotype groups including THC/CBD > 1 (Type I or THC-predominant), THC/CBD ≈1 (Type II or approximately equal proportions of THC and CBD), and THC/CBD < 1 (Type III or CBD-predominant); (**b**) the abundance (%) of Type I, Type II, and Type III plants measured for each location.

**Figure 4 plants-11-00129-f004:**
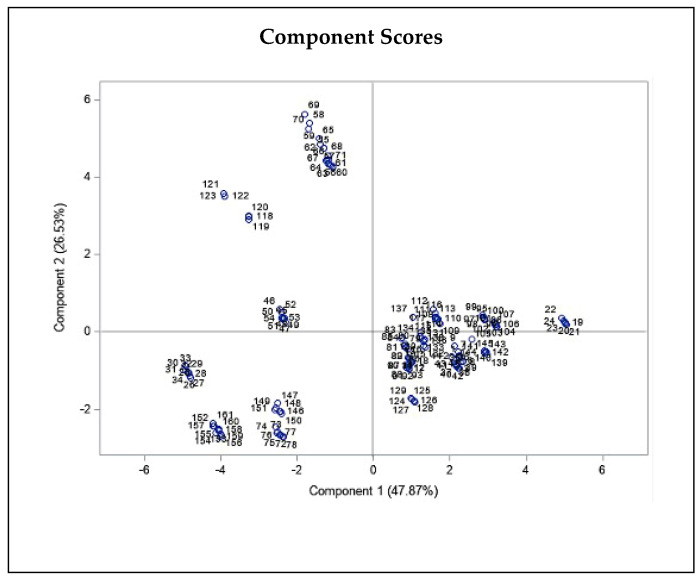
Principal component biplot consist of PCA scores of individuals (dots-first graphic) and loadings of variables (vectors-second graphic) of THC and CBD contents of 20 Iranian cannabis populations and geographical/climatic variables of regions of origin. PC1 and PC2 explain most of the variance in the original variables. Average annual temperature and latitude strongly influences PC1, while elevation and sunlight duration have more say in PC2. The loading plot shows THC and CBD contents positively correlate with mean, min, and max annual temperature and negatively correlate with elevation, latitude, and humidity statistics.

**Figure 5 plants-11-00129-f005:**
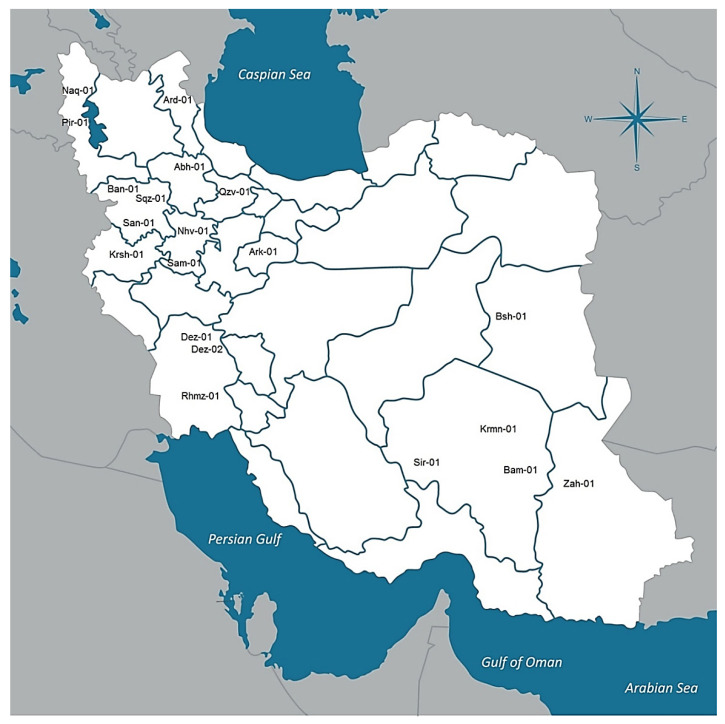
Geographical distribution of 20 distinct cannabis populations across selected regions of Iran.

**Figure 6 plants-11-00129-f006:**
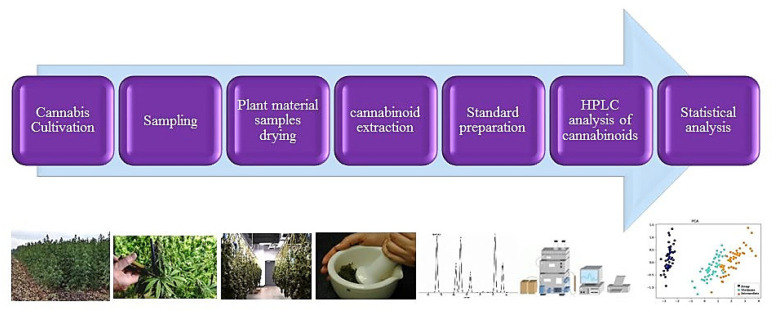
A workflow of cannabinoids assay and quantification.

**Table 1 plants-11-00129-t001:** Pearson correlation among geographical and climatic parameters with THC and CBD concentration in the studied cannabis plants.

Variables															
	THC	CBD	Longitude (E)	Latitude (N)	Elevation (m)	WS ^1^	Tmean ^2^	Tmin ^3^	Tmax ^4^	RHmean ^5^	RHmin ^6^	RHmax ^7^	SD ^8^	P ^9^	DP ^10^
**THC**	1	−0.053 ^NS^	0.118 ^NS^	−0.157 *	−0.408 **	−0.326 **	0.352 **	0.292 **	0.412 **	−0.103 ^NS^	−0.206 **	−0.057 ^NS^	−0.039 ^NS^	−0.114 ^NS^	0.305 **
**CBD**	−0.053 ^NS^	1	0.042 ^NS^	−0.239 **	−0.222 **	−0.040 ^NS^	0.314 **	0.311 **	0.287 **	−0.201 *	−0.171 *	−0.188 *	0.150 ^NS^	−0.028 ^NS^	0.149 ^NS^

** Correlation is significant at the 0.01 level; * Correlation is significant at the 0.05 level; ^NS^, non-significant. ^1^ Wind speed. ^2^ Average annual temperature. ^3^ Average annual minimum temperature. ^4^ Average annual maximum temperature. ^5^ Average annual relative humidity. ^6^ Average annual minimum humidity. ^7^ Average annual maximum humidity. ^8^ Sunshine duration. ^9^ Average annual precipitation. ^10^ Dew point.

**Table 2 plants-11-00129-t002:** Geographical data of native cannabis genotypes were collected from different selected regions of Iran.

Country	Province	Location Code	Sample Size	Elevation (m)	Longitude (E)	Latitude (N)	Annual Average Temperature (C)	Annual Rainfall (mm)	Seed to Harvest (Day)
Iran	West Azerbaijan	Pir-01	7	1572	45°08′	36°42′	13.28	640.61	50
Iran	Kurdistan	San-01	9	1464	46°99′	35°31′	14.00	375.10	209
Iran	Hamadan	Nhv-01	8	1666	48°25′	34°15′	14.72	385.29	198
Iran	Kurdistan	Ban-01	10	1503	45°53′	35°59′	14.26	660.88	206
Iran	Kurdistan	Sqz-01	7	1480	46°26′	36°24′	11.20	439.37	53
Iran	Kerman	Sir-01	6	1754	55°68′	29°43′	17.81	138.52	207
Iran	Ardabil	Ard-01	6	1339	48°29′	38°24′	9.12	273.67	165
Iran	Sistan and Balochistan	Zah-01	10	1352	60°86′	29°49′	19.30	73.58	208
Iran	Khuzestan	Rmhz-01	6	179	49°59′	31°27′	27.67	280.49	210
Iran	Markazi	Ark-01	9	1722	49°42′	34°04′	14.04	297.75	200
Iran	Qazvin	Qzv-01	10	1315	49°86′	36°47′	13.91	311.85	183
Iran	West Azerbaijan	Naq-01	7	1324	45°22′	36°57′	14.08	323.95	58
Iran	Zanjan	Abh-01	9	1543	49°02′	36°28′	12.39	301.31	179
Iran	Hamadan	Sam-01	6	1858	48°70′	34°20′	13.53	324.79	198
Iran	Kerman	Krmn-01	7	1761	56°58′	30°15′	17.01	123.43	185
Iran	Kermanshah	Krsh-01	7	1389	47°03′	34°19′	15.51	402.63	206
Iran	Kerman	Bam-01	10	1068	58°14′	29°09′	23.74	54.05	208
Iran	South Khorasan	Bsh-01	10	881	57°43′	34°03′	21.07	79.98	174
Iran	Khuzestan	Dez-01	9	144	48°42′	32°38′	24.56	389.40	206
Iran	Khuzestan	Dez-02	8	144	48°42′	32°38′	24.56	389.40	201

## Data Availability

Data are contained within the article or Appendix A.

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
