# Peer review of "THC and CBD Fingerprinting of an Elite Cannabis Collection from Iran: Quantifying Diversity to Underpin Future Cannabis Breeding"

_plants, 2022, doi:10.3390/plants11010129_

Round 1

Reviewer 1 Report

Cannabis and its  metabolites have long been the subject of extensive research and testing CBD has shown a wide range of pharmacological activities in several preclinical studies and clinical studies that have allowed its potential application in some cases of inflammation, cancer, cardiovascular diseases, epilepsy and neurodegenerative and psychiatric disorders.

Recent discoveries on the medicinal properties of cannabis and its industrial applications have given impetus to research into the structure of this plant's genome.

On the Iranian cannabis market, there is little information regarding the chemical characteristics of the cannabis varieties available; the AA. considered that THC / CBD concentration and ratio could be the crucial key points for starting a breeding program in the huge cannabis market.

The study is interesting, well written, organized and sufficiently detailed in such a way as to render the results inherent to the analytical method used for the determination of the biochemical profile of THC and CBD in HPLC of 161 wild pistillate cannabis plants from 20 regions of Iran. , easily usable.

The meaning that the AAs attribute to the results plays a fundamental role and leads to a reflective analysis that guides them towards their clinical and theoretical usefulness. Overall, the results showed a correlation between geography and climate of the site of origin, suggesting that both THC and CBD production were correlated with temperature and latitude, respectively, but surely more research and insights are needed.

The abstract allows you to know the main information about the study, objective, method, main results and conclusions

In the introduction we find the problem of the study clearly explained, its level of importance and its context. nevertheless, please cut the introduction to foscu only on the principal issue of the study.

In the conclusions that are not well balanced with respect to the discussion, the authors expressed the implications of the results of their study in terms of practical repercussions and future developments of the discipline, and plan to deepen the research, to broaden the sample. Please create a pbalance beween Discussion and Conclusion Sections in terms of lenght.

The scientifically qualitative bibliography deserves a further update with respect to the topic dealt with, there are numerous studies on the subject that are not mentioned.

It is advisable to consult the following studies

Brunetti P, Pichini S, Pacifici R, Busardò FP, Del Rio A. Herbal Preparations of Medical Cannabis: A Vademecum for Prescribing Doctors. Medicina (Kaunas). 2020 May 15;56(5):237. doi: 10.3390/medicina56050237.

Pacifici R, Marchei E, Salvatore F, Guandalini L, Busardò FP, Pichini S. Evaluation of long-term stability of cannabinoids in standardized preparations of cannabis flowering tops and cannabis oil by ultra-high-performance liquid chromatography tandem mass spectrometry. Clin Chem Lab Med. 2018 Mar 28;56(4):94-96. doi: 10.1515/cclm-2017-0758.

Roberta Pacifici, Emilia Marchei, Francesco Salvatore, Luca Guandalini, Francesco Paolo Busardò and Simona Pichini

Evaluation of cannabinoids concentration and stability in standardized preparations of cannabis tea and cannabis oil by ultra-high performance liquid chromatography tandem mass spectrometry. Clin Chem Lab Med. 2017 Aug 28;55(10):1555-1563. doi: 10.1515/cclm-2016-1060.

 Emilia Marchei, Roberta Tittarelli, Manuela Pellegrini, Maria Concetta Rotolo, Roberta Pacifici, Simona Pichini. Is "light cannabis" really light? Determination of cannabinoids content in commercial products

Ther Drug Monit2020 Feb;42(1):151-156. doi: 10.1097/FTD.0000000000000683

Brunetti P, Lo Faro AF, Pirani F, Berretta P, Pacifici R, Pichini S, Busardò FP. Pharmacology and legal status of cannabidiol. Ann Ist Super Sanita. 2020 Jul-Sep;56(3):285-291. doi: 10.4415/ANN_20_03_06.

Reviewer 2 Report

In the present manuscript, 20 populations of cannabis plants from 20 geographical regions of Iran were characterized in terms of their tetrahydrocannabinol (THC) and cannabidiol (CBD) content.

Considering that all the plants were cultivated under the same conditions (see Lines 351-353) I do not understand how the authors analysed the Genotype X Environment interaction in their tested hypothesis. Moreover, it is not clear how the meteorolical conditions of the regions of origin affected the chemical composition of the plants that were cultivated under the same conditions.

The exeriments where G X E interaction is tested usually employ multisite and/or multiyear experiments. In this case, all the genotypes were cultivated under the same conditions.   

Author Response

We would like to thank the reviewer for careful reading of the manuscript and for the helpful comments and constructive suggestions. Appropriated changes suggested by the reviewer 2, have been introduced to the manuscript. These changes are described below in a point-to-point manner (our responses are in red) and are highlighted in yellow within the document.

Response to Reviewer 2 Comments

Point 1: In the present manuscript, 20 populations of cannabis plants from 20 geographical regions of Iran were characterized in terms of their tetrahydrocannabinol (THC) and cannabidiol (CBD) content.

Considering that all the plants were cultivated under the same conditions (see Lines 351-353) I do not understand how the authors analysed the Genotype X Environment interaction in their tested hypothesis. Moreover, it is not clear how the meteorological conditions of the regions of origin affected the chemical composition of the plants that were cultivated under the same conditions.

The experiments where G X E interaction is tested usually employ multisite and/or multiyear experiments. In this case, all the genotypes were cultivated under the same conditions.   

Response 1: Authors thank the reviewer 2 for mentioning this. It is true that the experiments where G X E interactions are tested usually employ multisite or multiyear. In this research, we believe that female flower buds were sampled from 20 populations grew in different regions of Iran with different meteorological patterns to measure THC and CBD content of this collection. Moreover, we included the environment conditions of sampled area to assess the correlation of these climatic/geographical variables with THC and CBD content of studied individuals. In addition to female flowers, open-pollinated seeds from each location were sampled and grown in the research field of the University of Tehran for next studies including drought stress experiment and morphological and genomic evaluation studies. So, this sentence was point out our next studies towards our breeding goals. The paragraph was edited as follow:

“Twenty natural dioecious cannabis populations selected from the native cannabis gene pools across Iran (CGRC; www.medcannabase.org). These populations were located in distinct geographical and climatic regions (included cold and mountainous regions of the West and North-West, warm and wet regions of the South-West and warm and dry regions of the East and South-East areas) (Figure 5; Table 2, Supplementary Table S.1.). At least 10 female plants per population were used for phytochemical analysis if available (Table 2). In addition, seeds were collected from labelled local populations to grow for next studies. Geographical information was recorded and used to determine climatic parameters of sampled regions.”

Reviewer 3 Report

This study provides a deep insight into 446 THC and CBD profile of 20 natural dioecious cannabis populations from distinct geographical regions of Iran. The work is very interesting but I think that the study could go much forward and be more complete. However, I recommend the publication of these results, if the authors consider some aspects:

Abstract: “G x E interactions”? Please explain.

Line 63: Please replace “Now a days,” by “Nowadays,”

Line 95: The first figure the authors mentioned is Figure 5? Why not Figure 1? The same can be asked for Table 2.

In my opinion, the explanation of the division in Type I, Type II, Type III is to repeated over the manuscript.

On the PCA analysis, the authors do not identify clearly what are the PC1 and PC2 components. Also, can the authors add vertical and horizontal lines with a 0,0 intersection in the first graphic of Figure 4? In my opinion, it would help to interpret the results expressed on the second graphic in terms of quadrants.

Author Response

We would like to thank the reviewer for careful reading of the manuscript and for the helpful comments and constructive suggestions. Appropriated changes suggested by the reviewer 3, have been introduced to the manuscript. These changes are described below in a point-to-point manner (our responses are in red) and are highlighted in yellow within the document.

Response to Reviewer 3 Comments

Point 1: Abstract: “G x E interactions”? Please explain.

Response 1: Gene–environment interaction (or genotype–environment interaction or G x E or G×E) is well described in the literatures; we thought this might be enough to explain about it here. Genotype–environment interaction (in the literatures, it is written as G x E interactions) is when different genotypes respond to environmental variation in different ways. Environmental variation can be physical, chemical, biological, behavior patterns or life events. In this study, we investigated the correlation of THC and CBD contents of Iranian cannabis collection (genotypes) with climatic and geographical parameters (environment). In the “Abstract” it has been revised and highlighted in yellow in the revised version.

Point 2: Line 63: Please replace “Now a days,” by “Nowadays,”

Response 2: We have done your comment and “Now a days,” replaced with “Nowadays,” and highlighted in yellow in revised version of the manuscript.

Point 3: Line 95: The first figure the authors mentioned is Figure 5? Why not Figure 1? The same can be asked for Table 2.

Response 3: We thank reviewer 3 for this comment. This is correct. Actually, Figure 5 and Table 2 are related to the information of studied samples and their locations which should come in section “4.1. Genetic resources” of “Materials and Methods”. So according to the order of figures and tables in the text, the number of this figure is 5 and the number of this table is 2 (both are in Materials and Methods). Since both Figure 5 and Table 2 should be embedded in “Materials and Methods” and in order to keep the order of the figure and table numbers in the text, we decided to don’t address “Figure 5; Table 2” in line 95 (line 78 in revised version) and only address it in main place (“Materials and Methods”). So, “Figure 5; Table 2” in line 95 (line 78 in revised version) was deleted and this line was re-written and highlighted in yellow in section 2.1. of revised version of the manuscript. Also, “Figure 5; Table 2” was highlighted in section 4.1. of “Materials and Methods”.

 Point 4: In my opinion, the explanation of the division in Type I, Type II, Type III is to repeated over the manuscript.

Response 4: The authors accepted this comment and we have deleted the extra explanation of the division in Type I, Type II and Type III throughout the manuscript.

Point 5: On the PCA analysis, the authors do not identify clearly what are the PC1 and PC2 components. Also, can the authors add vertical and horizontal lines with a 0,0 intersection in the first graphic of Figure 4? In my opinion, it would help to interpret the results expressed on the second graphic in terms of quadrants.

Response 5: We thank reviewer 3 for this comment. We have performed PCA to figure out if and how 13 climatic/geographical variables correlate with chemical composition (THC and CBD). This bi-plot PCA consist of two graphics, first graphic show the PCA scores of all individuals (dots) and the second graphic show the PCA variables (vectors). The information about PC1 and PC2 describing the most of variance in the variables as well as the variables that influence PC1 and PC2 and the correlations among variables was clarified in the caption of this figure (Figure 4) and highlighted in yellow in revised version of the manuscript. Also, we have improved this figure by adding the vertical and horizontal lines with a 0,0 intersection in the first graphic.

Round 2

Reviewer 2 Report

The response of the authors did not sufficiently answer my comment.

Author Response

Response to Reviewer 2 Comment

Round 1

Point: In the present manuscript, 20 populations of cannabis plants from 20 geographical regions of Iran were characterized in terms of their tetrahydrocannabinol (THC) and cannabidiol (CBD) content.

Considering that all the plants were cultivated under the same conditions (see Lines 351-353) I do not understand how the authors analysed the Genotype X Environment interaction in their tested hypothesis. Moreover, it is not clear how the meteorological conditions of the regions of origin affected the chemical composition of the plants that were cultivated under the same conditions.

The experiments where G X E interaction is tested usually employ multisite and/or multiyear experiments. In this case, all the genotypes were cultivated under the same conditions.   

Round 2

Point: The response of the authors did not sufficiently answer my comment.

Response: The authors apologize to the reviewer 2 if the answer to this comment was not sufficient and clear. Now we have tried to do our best to answer this comment clearly.

In this study, the female flower buds were sampled from 20 different geographical locations of Iran with different climatic patterns. So, we have included Genotype X Environment interaction in this study as well. In first version of the manuscript (submitted first), one sentence which had mentioned by the reviewer 2 in the comment (“Seeds were grown in the research field of the University of Tehran with the geographical specifications including latitude: 36â—¦ and 2 min North, longitude: 50â—¦ and 11 min East and elevation: 1260 m above sea level”) had been brought in this manuscript by a mistake and we greatly appreciate reviewer 2 for catching this mistake and pointing out.  

The reason why this mistake happened is that the University of Tehran has started a mega-project on cannabis plant from ten years ago. During the last ten years, more than 100 accessions were collected from different geographical regions of Iran. During germplasm collection trip, fresh samples from the top inflorescence (for chemical screening- current study) and seeds (for future studies to perform drought stress, genome survey and breeding programs) were sampled from the visiting growing sites.

Current study is the first study on chemical screening (potent THC and CBD profile) of female flower buds of wild Iranian populations located in distinct geographical and climatic regions (Figure 5; Table 2; Supplementary Table S.1.). To establish this collection and extend our studies and perform drought stress, genome survey, morphological features evaluation and breeding programs, in addition to female flower buds for chemical composition analysing (current manuscript), open pollinated seeds were also collected from those sampled regions to cultivate in the field of University of Tehran for next studies. Since the authors were writing and completing the manuscripts related to those studies (not yet published) as well, this sentence was mistakenly copied to this manuscript from another manuscript. Now, this sentence has been deleted and the whole paragraph of section “4.1. Genetic resources” of “Materials and Methods” has been revised and highlighted in yellow in revised version of the manuscript.
